# Her Majesty’s Desert Throne: The Ecology of Queen Butterfly Oviposition on Mojave Milkweed Host Plants

**DOI:** 10.3390/insects11040257

**Published:** 2020-04-21

**Authors:** Steven M. Grodsky, Leslie S. Saul-Gershenz, Kara A. Moore-O’Leary, Rebecca R. Hernandez

**Affiliations:** 1Wild Energy Initiative, John Muir Institute of the Environment, University of California, Davis, 1 Shields Ave., Davis, CA 95616, USA; lsaulgershenz@ucdavis.edu (L.S.S.-G.); rrhernandez@ucdavis.edu (R.R.H.); 2Department of Entomology and Nematology, University of California, Davis, 1 Shields Ave., Davis, CA 95616, USA; 3Center for Population Biology, University of California, Davis, 1 Shields Ave., Davis, CA 95616, USA; kara_moore-oleary@fws.gov; 4Department of Land, Air & Water Resources, University of California, Davis, 1 Shields Ave., Davis, CA 95616, USA

**Keywords:** *Danaus*, host plant, insect–plant interactions, Mojave Desert, Mojave milkweed, oviposition, queen butterfly

## Abstract

Butterfly–host plant relationships can inform our understanding of ecological and trophic interactions that contribute to ecosystem function, resiliency, and services. The ecology of danaid–milkweed (Apocynaceae) host plant interactions has been studied in several biomes but is neglected in deserts. Our objective was to determine effects of plant traits, seasonality, and landscape-level host plant availability on selection of Mojave milkweed (*Asclepias nyctaginifolia* A. Gray) by ovipositing monarch butterflies (*Danaus plexippus*
*plexippus*) and queen butterflies (*Danaus gilippus thersippus*) in the Californian Mojave Desert. We surveyed all known Mojave milkweed locations in the Ivanpah Valley, California (*n* = 419) during early, mid-, and late spring in 2017. For each survey, we counted monarch and queen butterfly eggs on each Mojave milkweed plant. We also measured canopy cover, height, volume, and reproductive stage of each Mojave milkweed plant. We counted a total of 276 queen butterfly eggs and zero monarch butterfly eggs on Mojave milkweed host plants. We determined that count of queen butterfly eggs significantly increased with increasing Mojave milkweed canopy cover. Additionally, count of queen butterfly eggs was: (1) greater on adult Mojave milkweed plants than on juvenile and seedling plants and greater on juvenile Mojave milkweed plants than on seedling plants; and (2) greater during early spring than mid-spring—we recorded no eggs during late spring. Based on aggregation indices, queen butterfly eggs occurred on Mojave milkweed plants in a nonrandom, clustered pattern throughout the Ivanpah Valley. We provide the first evidence of trophic interactions between queen butterflies and Mojave milkweed at multiple spatial scales in the Mojave Desert, suggesting that conservation and management practices for both species should be implemented concurrently. Given its role as an herbivore, pollinator and prey, the queen butterfly may serve as a model organism for understanding effects of anthropogenic disturbance (e.g., solar energy development) on “bottom-up” and trophic interactions among soils, plants and animals in desert ecosystems.

## 1. Introduction

Desert ecosystems maintain abundant insect–plant interactions, supporting trophic and symbiotic relationships, shaped by coevolution of species in environments with highly variable abiotic conditions and limited resources [1,2]. Classic examples of insect–plant interactions in deserts include biblical plagues of phytophagous locusts, ant granivory, and native bee pollination [3,4,5]. Research elucidating the influence of extrafloral nectaries on mutualisms between cacti and ants further illustrates the depth of known and yet to be discovered insect–plant interactions in desert ecosystems [6]. Many insects, including butterflies, use a suite of desert plants for food (e.g., leaves, nectar), sources of chemicals for mating and defense, and oviposition sites throughout their life histories [7,8,9].

Among all butterfly–host plant interactions, the story of the monarch butterfly (*Danaus plexippus plexippus*) and its milkweed (Apocynaceae) host plants is most prevalent in science and society today [10]. Recent modeling efforts of habitat suitability for western monarch butterflies based on multiple criteria, including availability of certain milkweed host plants, showed that deserts of the southwestern United States may provide a large area of suitable habitat for monarch butterflies [11]. The western population of monarch butterflies has declined in number by >95% since the 1980s, surpassing estimated declines of the larger eastern population of the species [12]. Indeed, the monarch butterfly is currently under consideration for federal protection under the United States Endangered Species Act. In addition to the monarch butterfly, its congener the queen butterfly (*Danaus gilippus thersippus*) also occurs in deserts of the southwestern United States [13]. Although the tritrophic interactions among milkweed plants, monarch and queen butterflies, and predators represent a well-documented paradigm in ecology [14], few studies have addressed these interactions in natural desert environments relative to other biomes. Further, many studies on monarch butterfly–milkweed interactions have been conducted in laboratory settings [15], and studies that have addressed monarch butterfly use of desert milkweeds in the field have focused on a subset of milkweed species (e.g., *Asclepias eriocarpa*, *A. erosa*) using lab-reared caterpillars (e.g., [16]). In comparison to the western monarch butterfly, virtually nothing is known about queen butterfly ecology in the western United States [17].

Of all the desert milkweed species potentially used as host plants by monarch and queen butterflies, perhaps none capture the essence of the desert environment, in terms of distribution and life history, more than the aptly named Mojave milkweed (*Asclepias nyctaginfolia* A. Gray). Mojave milkweed is an herbaceous, perennial desert plant sporadically distributed in the southwestern United States [18]. As a seasonally iteroparous plant, Mojave milkweed emerges in spring and fall from underground tubers, and it reproduces when soil moisture is sufficient for fruit production [19]. The plant has broad, oval to lance-shaped leaves that are thick and fleshy. We observed pubescent Mojave milkweed leaves in the field (S. M. Grodsky, pers. obs.); however, a previous study designated Mojave milkweed leaves grown from seed in a sphagnum peat moss-based soil as glabrous [20]. The same researchers measured phytochemical concentrations in 24 milkweed species distributed throughout the United States and determined that Mojave milkweed has high concentrations of cardenolides, quercetin glycosides, and total phenolics relative to northern and eastern *Asclepias* species [20]. Milkweed species sympatric with Mojave milkweed, including *A. asperula* and *A. californica*, also exhibited above-average phytochemical concentrations, indicating that desert *Asclepias* species may have adapted more potent chemical defenses in resource limited environments [20,21]. 

Monarch and queen butterfly oviposition on Mojave milkweed may fluctuate seasonally as a spatiotemporal function of variable resource availability in desert ecosystems. Multivoltine butterflies may exhibit adaptive seasonal plasticity in response to temperature, for example, that may influence life cycle regulation [22]. Competition between congeners like the monarch and queen butterfly may occur in deserts, in part, as a function of intra- and inter-seasonal rainfall; for example, *Papilio indra fordi* Comstock and Martin and *P. rudkini* Comstock directly competed for foodplants following a heavy rainfall year but not during dry years in the Californian Mojave Desert [7]. Milkweed host plant availability in desert landscapes also may vary seasonally as a result of precipitation and herbivory [18]. Monarch butterfly oviposition preference has been found to depend on both the size of milkweed patches and density of milkweed plants within patches [23], both of which may be affected by seasonal variables in desert ecosystems.

Because larvae of butterflies often have little opportunity to drastically alter their developmental location, larval survival as dictated by available nutrition, exposure to the elements, and susceptibility to predation may be largely determined by selection of Mojave milkweed host plants by ovipositing monarch and queen butterflies [16,24]. Although butterfly larvae may be able to move among available hosts in some ecosystems, the interspaces between plants and aridity in deserts may preclude such larval movement as a means by which to increase survival for some species [25]. Further, variability in the abundance of Mojave milkweed host plants as a function of anthropogenic activity, for instance, may influence adaptation in oviposition strategies [26]. Therefore, understanding Mojave milkweed host-plant selection by monarch and queen butterflies may shed light on the management and conservation of these species in the Mojave Desert today.

The goal of this study was to address gaps in knowledge regarding monarch and queen butterfly-milkweed host plant interactions in a desert ecosystem using field-collected, empirical data rather than laboratory experiments, modeling exercises, and single-event natural history notes. To this end, our objective was to determine relationships among plant traits, seasonality, and landscape availability of Mojave milkweed and the count of monarch and queen butterfly eggs oviposited on Mojave milkweed plants in the Mojave Desert. We hypothesized that monarch and queen butterflies used Mojave milkweed as a host plant in the Mojave Desert because Mojave milkweed is endemic to the region and it is known to contain phytochemicals available for sequestration by monarch and queen caterpillars.

## 2. Materials and Methods

We conducted our study in the Ivanpah Valley of the Mojave Desert, San Bernardino County, California, USA (Figure 1). The Ivanpah Valley is characterized geologically by piedmonts, intersecting active and inactive alluvial fans and channels, and terminal playas [27]. The climate in the Ivanpah Valley is hot and dry, with summer midday temperatures often exceeding 40 °C and annual precipitation averaging ~13.50 cm mostly during the winter and summer monsoon seasons. The vegetation community in the Ivanpah Valley is classified as creosote desert scrub. Creosote desert scrub covers over 6.5 million ha of the Californian Mojave Desert, an area roughly 400,000 ha larger that the state of West Virginia (USA). Creosote desert scrub is characterized by evenly distributed shrubs, including creosote (*Larrea tridentata*) and white bursage (*Ambrosia dumosa*), with other desert plant species occurring either under shrubs [e.g., beavertail cactus (*Opuntia basilaris*)] or in the interspaces between them [e.g., Mojave yucca (*Yucca schidegera*)] (Figure 2).

From late April to early June 2017, we studied the ecology of western monarch and queen butterfly oviposition on Mojave milkweed host plants at all known Mojave milkweed locations in the Ivanpah Valley (*n* = 419). We based known Mojave milkweed locations in the Ivanpah Valley on GPS coordinates generated from field surveys of the species conducted by K. A. Moore-O’Leary and S. M. Grodsky, respectively, over the last decade. We surveyed Mojave milkweed locations in early spring (27 April–4 May), mid-spring (15 May–19 May), and late spring (30 May–1 June) of 2017 to determine relationships between seasonality and monarch and queen butterfly during their breeding seasons. We observed no herbivory on Mojave milkweed plants in early spring, indicating that butterfly eggs had yet to develop into caterpillars and that, while certainly possible, oviposition events prior to the first spring survey were few. During each survey, we characterized traits of each Mojave milkweed plant by taking the following measurements: (1) length along horizontal plane (cm) (hereafter “canopy cover”); (2) width along horizonal plane (cm); and (3) height along vertical plane (cm) (hereafter “height”). For each plant measured during each survey, we assigned a life stage (seedling = ≤8-cm canopy cover and non-reproductive, juvenile = ≥8-cm canopy cover and non-reproductive, and adult = any plant with reproductive structures present) and calculated plant volume (cm^3^) as length*width*height. Concurrently, we counted the number of monarch and queen butterfly eggs on all leaves of each Mojave milkweed plant present during each survey. We used a 10× loupe to differentiate between monarch and queen butterfly eggs to species based on egg morphology (e.g., [28]). We conducted the study following a period of relatively high preemergence and growing season rainfall (winter/spring rainfall = 9.98 cm) in the Ivanpah Valley.

We developed a global Poisson generalized linear model (GLM) to determine relationships among Mojave milkweed traits and seasonality and count of queen butterfly eggs oviposited on Mojave milkweed host plants. We used cumulative counts of queen butterfly eggs recorded on each plant during each survey as the dependent variable and included season (i.e., early, mid-, and late spring), Mojave milkweed life stage, Mojave milkweed canopy cover, Mojave milkweed height, and Mojave milkweed volume as independent variables in the model (Table 1). Prior to analyses, we tested for correlation between Mojave milkweed life stage and Mojave milkweed volume by assessing the fit of a logistic regression with volume as predictor for life stage and determined that there was no correlation, which was corroborated by field observations (e.g., reproductive structures on small plants; large, non-reproductive plants; SMG, personal observation). We performed a likelihood ratio test on the GLM to determine significant covariate effects. We conducted post-hoc Tukey’s pairwise comparisons of categorical variables (i.e., season, life stage) egg count means using general linear hypothesis testing (glht function, single step method) with a Bonferroni adjustment in the R package “multcomp” [29]. We reported beta coefficients of the GLM generated from the R package “reghelper” [30]. We set α = 0.05. We calculated Fischer’s index of aggregation and Lloyd’s index of patchiness for queen butterfly eggs oviposited on Mojave milkweed plants throughout the Ivanpah Valley, using the R package “epiphy” [31]. We used the count of queen butterfly eggs per Mojave milkweed plant cumulatively recorded over the entire study period for aggregation analyses; both Fisher’s and Lloyd’s index accommodate count data. We summarized qualitative field observations elucidating queen butterfly–Mojave milkweed interactions with regard to oviposition.

## 3. Results

We counted a total of 276 queen butterfly eggs and zero monarch butterfly eggs on Mojave milkweed host plants in the Ivanpah Valley of the Mojave Desert during the spring of 2017. We counted the majority of queen butterfly eggs (88%) and Mojave milkweed plants (75%) in early spring of 2017. We documented that queen butterfly oviposition and availability of Mojave milkweed plants peaked in early spring and continuously declined through mid-spring to late spring of 2017 (Figure 3). During the study period, we found queen butterfly eggs on 115 individual Mojave milkweed plants, comprising ~27% of total Mojave milkweed plants in the region. 

We detected a significant relationship between Mojave milkweed canopy cover and count of queen butterfly eggs (DF = 1, LRT = 15.54, Prχ^2^ = 8.07 × 10^−5^); the count of queen butterfly eggs significantly increased with increasing Mojave milkweed canopy cover (Figure 4). We found a negative relationship between both Mojave milkweed height and volume and count of queen butterfly eggs, although the effect of neither variable was statistically significant (Figure 4). We also documented a significant relationship between Mojave milkweed life stage and count of queen butterfly eggs (DF = 2, LRT = 48.49, Prχ^2^ = 2.96 × 10^−11^). The count of queen butterfly eggs was greater on adult Mojave milkweed plants (mean canopy cover = 30.56 cm; SE = 1.85) than on juvenile (mean canopy cover = 17.32 cm; SE = 0.59) and seedling (mean canopy cover = 5.31 cm; SE = 0.20) Mojave milkweed plants; the count of queen butterfly eggs was greater on juvenile Mojave milkweed plants than on seedling Mojave milkweed plants (Figure 5). We also documented a significant relationship between seasonality and count of queen butterfly eggs (DF = 2, LRT = 37.01, Prχ^2^ = 9.19 × 10^−9^). We counted more queen butterfly eggs on Mojave milkweed host plants during early spring than during mid-spring; we counted no queen butterfly eggs during late spring. Based on comparison of standardized coefficients, the seedling life stage (β = −1.00, SE = 0.18) was the strongest predictor of count of queen butterfly eggs relative to other significant independent variables, such as canopy cover (β = 0.37, SE = 0.09) (Table 1). We determined that queen butterfly eggs occurred on Mojave milkweed in a nonrandom, aggregated pattern throughout the Ivanpah Valley (Fischer’s index of aggregation = 4.07; Lloyd’s index of patchiness = 7.20) (Figure 1).

On rare occasions, we observed queen butterfly females actively ovipositing on Mojave milkweed host plants. On each occasion, the female oviposited a single egg on the abaxial side of a Mojave milkweed leaf after some sporadic flying maneuvers above the plant (Figure 6a). In most cases, we found one queen butterfly egg laid on the abaxial (underside) surface of a Mojave milkweed leave (Figure 6b). However, we encountered a fair number of queen butterfly eggs laid on adaxial (upper side) surface of Mojave milkweed leaves (Figure 6c). We encountered several Mojave milkweed plants, many of which were reproductive, with multiple queen butterfly eggs laid on them (Figure 6c). The highest number of queen butterfly eggs we found on an individual Mojave milkweed host plant was 19. We also recorded cases in which queen butterfly eggs were laid quite near to one another on the same Mojave milkweed leaf (Figure 6d).

## 4. Discussion

Our study is the first to document queen butterfly oviposition on Mojave milkweed host plants in the Californian Mojave Desert. To our knowledge, every account of queen butterfly oviposition on milkweed in this region to date is based on field observations of one larva on specific milkweed species; these include sightings of queen butterfly caterpillars on *Asclepias albicans*, *A. asperula*, *A. erosa*, *A. fascicularis*, *A. incarnata*, *A. speciosa*, *A. subulata*, *Funastrum hirtellum*, and *Sarcostemma cynanchoides* [13,32,33]. In general, early lepidopterists have suggested that vine-like milkweeds, including *F. hirtellum* and, to a lesser extent, *S. cynanchoides*, are the primary larval hosts of queen butterflies in the Californian Mojave Desert [28]. Since 1925, five of the eleven Asclepiad species known to occur in the region of our study site have been verified as queen butterfly host plants in the literature preceding this paper [34]. Given the scarcity of Mojave milkweed relative to other milkweed species in the Mojave Desert [35], coupled with the high count of queen butterfly eggs recorded on Mojave milkweed plants in the Ivanpah Valley, the queen butterfly may preferentially select Mojave milkweed host plants. However, we have no comparative measures of queen butterfly oviposition on other milkweed species in the study area.

Our results indicate that western monarch butterflies—a congener of the queen butterfly—may not use Mojave milkweed as a host plant in the Mojave Desert, possibly due to factors such as distribution and interspecific competition. First, monarch butterflies may not breed in the study area, whereas queen butterflies are known to breed throughout the Californian Mojave Desert [13]. Observations and museum records indicate that monarch butterflies in the southwestern United States are mostly distributed along rivers and select habitat in riparian corridors during autumn migration [36]. Queen butterflies may outcompete monarch butterflies for milkweed host plants in the Mojave Desert. Seminal studies by the late lepidopterist Lincoln Brower suggested that interspecific competition occurs between monarch and queen butterflies (*D. g. berenice*) in Florida, USA, as evident by egg cannibalism among the congeners and allopatry during annual monarch migrations [37,38]. On the other hand, wet conditions preceding the growing season at the study site may have led to an abundance of diverse milkweed host plants on the landscape; as such, it is possible that queen and western monarch butterflies simply selected different milkweed host plants for oviposition during the study period. Monarch butterflies are thought to require one and half times more larval food than queen butterflies [38], which may indicate that monarchs select larger, more voluminous milkweed species occurring in the Mojave Desert (e.g., *A. erosa*), whereas the smaller, lower-lying Mojave milkweed sufficiently supports queen butterfly larvae in the same region.

We can draw from the plethora of hypotheses regarding monarch butterfly oviposition on milkweed hosts (see [39]) to postulate female queen butterfly selection of individual Mojave milkweed host plants for oviposition. For all oviposition decisions exhibited by herbivorous insects, the classic preference-performance hypothesis posits that females should prefer to oviposit on plants that facilitate high offspring performance [40]. However, sequestering insects like monarch and queen butterflies may be an exception to the preference-performance hypothesis because their oviposition preference for highly defended offspring can add layers of chemical and ecological complexity to oviposition decisions that span beyond larval growth alone [39]. Milkweed host-plant selection by monarch and queen butterflies may be further complicated by the fact that cardenolide production is influenced by phenotypic plasticity among milkweeds in response to biotic and abiotic environmental conditions and is subject to natural selection by herbivores [14]. *Plant growth*—While previous studies have indicated that monarch butterflies preferentially oviposit on taller, more voluminous milkweed plants in temperate climates (e.g., [41,42,43]), queen butterfly oviposition on Mojave milkweed host plants in the Mojave Desert increased with canopy cover but not with height or volume. Queen butterfly selection of Mojave milkweed with high canopy cover may support the preference-performance hypothesis because as a low-lying and sprawling species, Mojave milkweed with high canopy cover likely provides more larval food than taller specimens, which can be lanky with relatively sparse leaves due to competition with surrounding plants (S.M. Grodsky, personal observation). *Plant age*—Some studies suggest that monarch butterflies prefer to oviposit on younger milkweed plants (e.g., [42]); queen butterflies exhibited the opposite preference for successively older Mojave milkweed host plants. Our preliminary analyses reveal that Mojave milkweed life stage is not correlated with the volume of Mojave milkweed, which suggests that chemical defense rather than larval food availability influences queen butterfly oviposition preference for individual Mojave milkweed plants; however, chemical analyses of the plant material would be required to validate this reasoning. *Cardenolides*—Several studies on monarch oviposition decisions indicated that number of monarch eggs may increase at intermediate cardiac glycoside levels [39,44]. Researchers determined that adult queen butterflies captured in xeric sites in Florida (USA) had significantly lower cardenolide concentrations than those captured in hydric sites [45], potentially indicating that queen butterflies in aridlands may prefer individual Mojave milkweed plants with lower concentrations of cardenolides than those with higher concentrations. It is possible that cardenolide production in Mojave milkweed is physiologically linked to life stage such that tradeoffs exist between energy input for plant defense and that for reproduction in adult plants [46], thereby creating intermediate cardiac glycoside levels thought to be preferred by monarch and queen butterflies.

Our field observations provide unique insights into the oviposition behavior of queen butterflies in the Mojave Desert that contradict some existing hypotheses on monarch and queen butterfly oviposition preference. Monarch caterpillars are well-known to be negatively impacted by intraspecific density due to intraspecific egg cannibalism and competition [47,48]. Meanwhile, queen butterflies have been estimated to exhibit egg cannibalism rates twice those of monarch butterflies [37]. Several times we observed queen butterfly eggs laid in close proximity on the same Mojave milkweed plant and, in some cases, directly adjacent to one another on the same Mojave milkweed leaf, which may indicate intraspecific density does not necessarily impact oviposition site selection by female queen butterflies in the Mojave Desert. Furthermore, many observations of monarch oviposition indicate that females lay eggs singly on the abaxial side of milkweed leaves (e.g., [42]). Our observations of queen butterfly oviposition on Mojave milkweed also signify that females lay a single egg on one host plant; however, we encountered quite a few cases in which females oviposited their eggs on the adaxial side of Mojave milkweed leaves, which likely makes eggs more visible to predators than those concealed on the abaxial side of leaves. Adult male queen and monarch butterflies transfer pyrrolizidine alkaloid (PA) that they sequester from plants to females during mating, which, in turn, transfer the PA to their eggs as a chemical defense against predation; indeed, the male courtship pheromone danaidone is derived from sequestered PA and may function to advertise potential for paternal allocation of PA to eggs as a measure of fitness [49]. Under the assumption that queen butterfly eggs are more vulnerable to predation on the exposed side of Mojave milkweed leaves than the abaxial side, our observations may suggest that females of the species pass sufficient chemical defenses against predation onto their eggs (e.g., PA) to negate the need for egg concealment.

The western queen butterfly has potential to serve as a model organism for better understanding ecological interactions in desert ecosystems of the United States and Mexico. Most hypotheses pertaining to the physical, chemical, and ecological relationships between the monarch butterfly and milkweed host plants in temperate climates can be tested for queen butterflies in deserts. Queen butterflies are herbivores as caterpillars, pollinators as adults, and potential prey throughout their life cycle; as such, studies on the species can shed light on a variety of trophic interactions to inform “bottom-up” ecological mechanisms in deserts ecosystems [50]. Invertebrates are considered excellent ecological indicators in a variety of ecosystems [51,52]. Because queen butterflies are inextricably tied to desert plants for reproduction (e.g., PA) and food, they may be especially useful indicators for alterations to sensitive desert plant communities caused by anthropogenic land-use and land-cover change. Further, queen butterflies can elucidate effects of these changes in deserts at multiple spatial scales. As seen with our study, both individual Mojave milkweed traits and Mojave milkweed distribution at the landscape level can affect queen butterfly oviposition in the Mojave Desert.

Desert ecosystems in the southwestern United States have been subjected to a barrage of anthropogenic disturbances, ranging from cattle grazing to nuclear bomb testing to utility-scale solar energy development [53,54], which can affect species-species and species-process interactions. Interactions among Mojave milkweed, the western queen butterfly, and solar energy infrastructure in deserts may inform the ecological mechanisms behind ecosystem response to solar energy development. For example, cardenolide content in Mojave milkweed may decrease in response to shade cast by solar panels [14], which, in turn, may affect the chemical defenses and susceptibility to parasitism of queen butterflies. Similarly, Mojave milkweed is a desert specialist potentially maladapted to anthropogenic disturbance like solar energy development [18,55]; reductions in the availability of Mojave milkweed host plants due to solar energy development may affect queen butterflies and the ecosystem services they provide.

## 5. Conclusions

We provide the first evidence of trophic interactions between queen butterflies and Mojave milkweed at multiple spatial scales in the Mojave Desert, suggesting that conservation and management practices for both species should be implemented concurrently. Our results indicated that patterns of queen butterfly oviposition on milkweed host plants may be unique in deserts relative to more researched biomes; these patterns may specifically inform trophic interactions in desert ecosystems. Although some researchers caution against the use of “substitute” species (e.g., [56]), the queen butterfly may serve as a surrogate species for understanding the aridland ecology of imperiled western monarch butterflies, especially if climate change and habitat loss alter the current distribution of western monarchs in the southwestern United States. Given the precipitous decline in the western population of the monarch butterfly, abundance of the western queen butterfly also may be decreasing in response to some of the same stressors impacting its congener. Conservation efforts for the queen butterfly may be enhanced by knowledge of its host plant interactions in deserts. The queen butterfly may be a useful indicator species for studying effects of solar energy development and other anthropogenic disturbances on “bottom-up” and trophic interactions in desert ecosystems.

## Figures and Tables

**Figure 1 insects-11-00257-f001:**
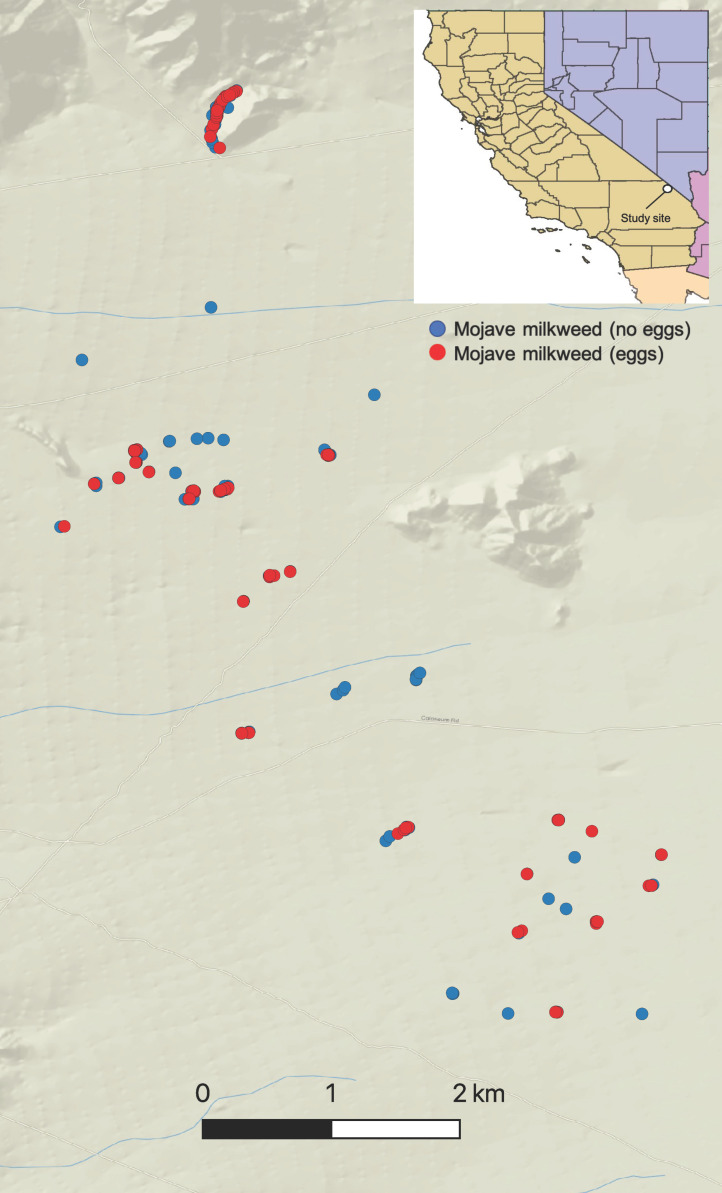
Distribution of Mojave milkweed plants without queen butterfly eggs (Mojave milkweed—no eggs, *n* = 304) and with at least 1 queen butterfly egg (Mojave milkweed—eggs, *n* = 115) recorded throughout the spring season, 27 April–1 June 2017, Ivanpah Valley, Mojave Desert, California, USA. Given the clustering of Mojave milkweed populations, individual Mojave milkweed points are not all visible. This map includes presence/absence of queen butterfly eggs on Mojave milkweed only, whereas aggregation analyses used egg count data per plant. Inset: Location of study site (white dot) in the United States. Map by Steve Grodsky.

**Figure 2 insects-11-00257-f002:**
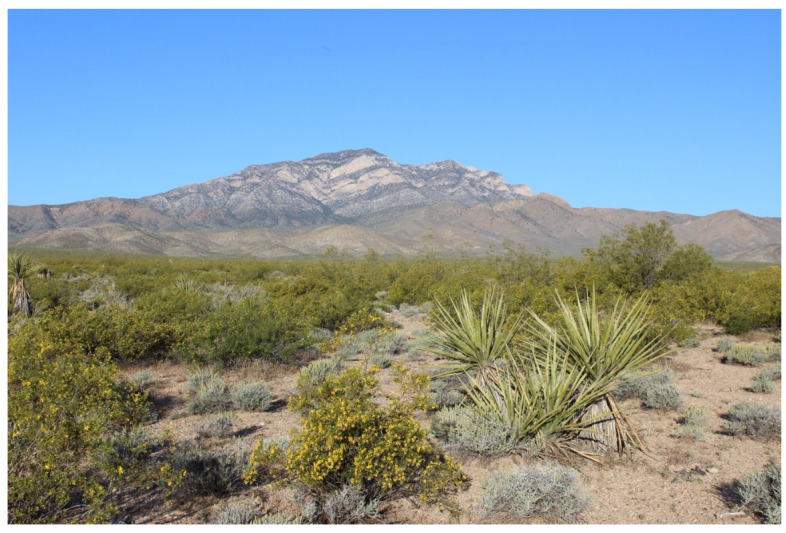
Creosote desert scrub vegetation community during spring in the Ivanpah Valley, Mojave Desert, California, USA. Photograph credit: Steve Grodsky.

**Figure 3 insects-11-00257-f003:**
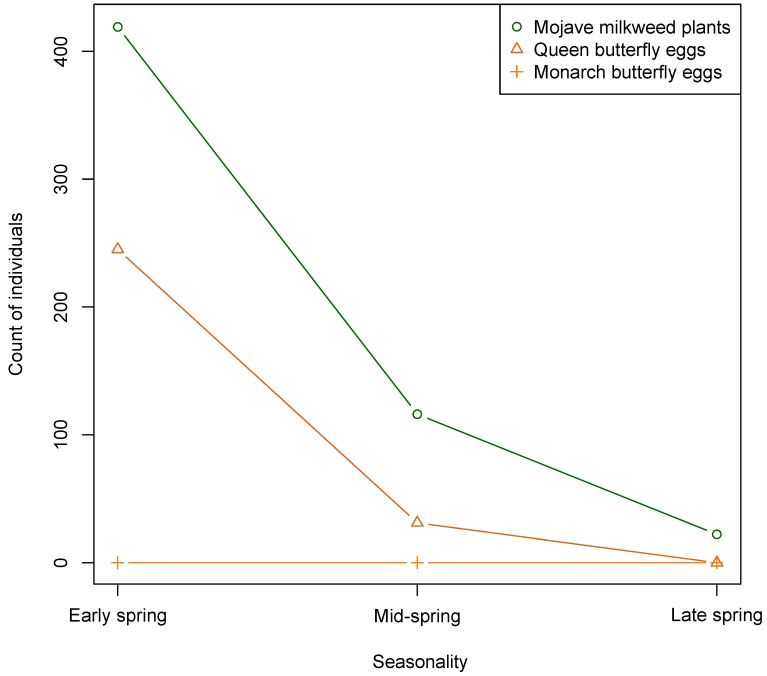
Count of monarch and queen butterfly eggs and Mojave milkweed plants recorded throughout the spring season, 27 April–1 June 2017, Ivanpah Valley, Mojave Desert, California, USA.

**Figure 4 insects-11-00257-f004:**
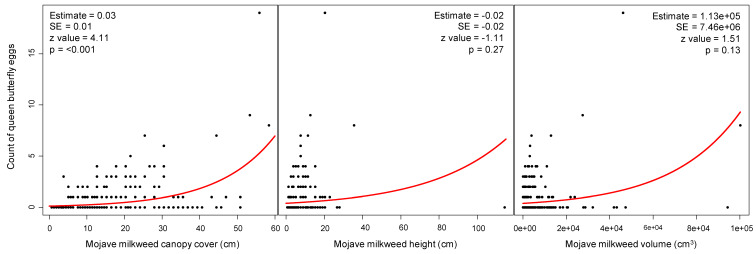
Effects of Mojave milkweed traits on the count of queen butterfly eggs recorded during the spring season, 27 April—1 June 2017, Ivanpah Valley, Mojave Desert, California, USA. Black dots indicate egg count data points and red lines indicate response curves based on the generalized linear model. We set α = 0.05

**Figure 5 insects-11-00257-f005:**
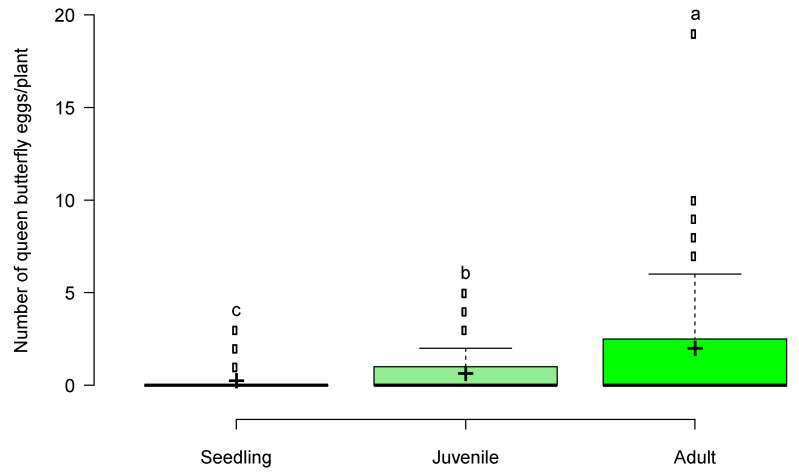
Box plot of queen butterfly oviposition on seedling, juvenile, and adult Mojave milkweed host plants documented during the spring season, 27 April–1 June 2017, Ivanpah Valley, Mojave Desert, California, USA. Different letters indicate significantly different pairwise caparisons of mean egg counts (plus signs) between life stages. We set α = 0.05

**Figure 6 insects-11-00257-f006:**
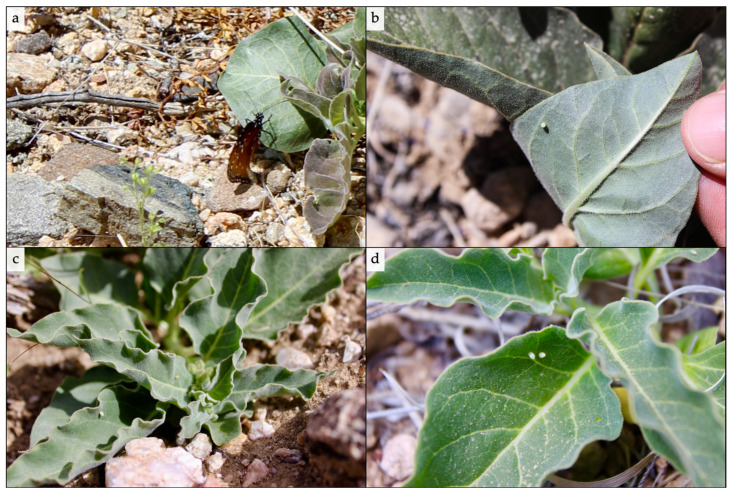
Queen butterfly oviposition on Mojave milkweed host plants during the breeding season, 27 April–1 June 2017, Ivanpah Valley, Mojave Desert, California, USA. (**a**) Adult queen butterfly, with abdomen under a leaf, ovipositing on a low-lying, young Mojave milkweed; (**b**) Commonly encountered egg placement on abaxial (underside) surface of Mojave milkweed leaf; (**c**) Three eggs on three leaves of an adult Mojave milkweed, note unopened flower buds behind front leaves of plant; (**d**) Two individual eggs laid adjacent to one another on the adaxial (upper side) surface of a Mojave milkweed leaf. Photographs by Steve Grodsky.

**Table 1 insects-11-00257-t001:** Standardized coefficients of the global Poisson generalized linear model developed to determine relationships between count of queen butterfly eggs oviposited on Mojave milkweed host plants and covariates

Covariate	β	Standard Error	z Value	*p* Value
*Seasonality* ^1^				
Early spring	0.51	0.10	4.42	<0.001
Mid-spring	−0.48	0.11	−4.46	<0.001
*Life stage*				
Seedling	−1.00	0.18	−6.11	<0.001
Juvenile	−0.54	0.09	−5.83	<0.001
Adult	0.31	0.05	5.94	<0.001
*Plant Traits*				
Canopy cover	0.37	0.09	4.03	<0.001
Height	−0.18	0.15	−1.19	0.23
Volume	0.11	0.07	1.63	0.10

^1^ We recorded no queen butterfly eggs in late spring.

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
