# Peer review of "Her Majesty’s Desert Throne: The Ecology of Queen Butterfly Oviposition on Mojave Milkweed Host Plants"

_insects, 2020, doi:10.3390/insects11040257_

Round 1

Reviewer 1 Report

This is a well written and carefully prepared manuscript, and was generally a pleasure to read.  Given the increasing interest in the monarch, it’s refreshing to see some work with the queen.  Mostly minor textual suggestions below and a few citations that might be useful to you.

Line 18.  It seems like weird word choice to say that butterfly-host plant relationships “inform” ecological and trophic interactions.  I think you mean that they inform our understanding of ecological and trophic interactions.

45.  Another hyper-picky word-choice comment.  Do insect-plant interactions really “span various trophic levels” or do they support various various trophic levels (and note that your examples in the next sentence are all directly plant-insect not multi-trophic).

87-88.  That seems unnecessary or can at least be re-worded to sound less trivial.  Of course phytochemical concentrations vary within populations (can you imagine a world in which they didn’t?).

180.  I would recommend putting all model results in a table.  That will make it easier for the reader to see how many and which exactly factors were included in models.

183.  Who cares if it jumps the arbitrary 0.05 barrier?  Tell us about the parameter estimate and associated uncertainty.  Is it interesting and biologically meaningful or not? 

187.  Typo: I think you mean “are” greater on juvenile plants.

193-194.  This is fine as far as it goes.  It would have been interesting to directly account for spatial autocorrelation through the Moran’s eigenvector mapping approach that takes your lat/longs and generates linear covariates that can be directly incorporated into models.  I wouldn’t insist that you do this (it would be a large change to your methods), but it’s also pretty easy to do in the adespatial package and you might learn something interesting (more than simply saying that there’s an aggregated pattern).

197.  I admit that I had to look up ab- and adaxial.  I don’t see how there’s anything gained vs the more commonly used upper and underside of the leaf. If you do go with the Latin, at least define it the first time you use it.

Figure 3.  All of the text on these figures needs to be made larger as it’s not going to come out well on a journal page.

Figure 4.  Although significant, are the differences here biologically meaningful given different volumes of plant material in different milkweed stages?  A bit more discussion would be useful, and this is also where it would be nice for the reader to be able to refer to a table with all covariates and standardized coefficients in one place.

226-228.  You might be interested to check out this collection of records that includes some from the Mojave.  Although it’s an obscure journal, I believe google scholar will pull up a pdf:

Austin GT, Leary PJ. Larval Hostplants Of Butterflies In Nevada. Holarctic Lepidoptera. 2008 Aug 1;12(1-2).

235.  I think you mean “may preferentially select” (after all, of course they select them).

241-242.  You might be interested in the habitat model results in this paper:

Dilts, T., Steele, M., Engler, J.D., Pelton, E.M., Jepsen, S.J., McKnight, S., Taylor, A.R., Fallon, C.E., Black, S.H., Cruz, E.E. and Craver, D.R., 2019. Host plants and climate structure habitat associations of the western monarch butterfly. Frontiers in Ecology and Evolution, 7, p.188.

Reviewer 2 Report

Review of Manuscript ID: insects-767291

Title: Her Majesty’s desert throne: The ecology of queen butterfly oviposition on Mojave milkweed host plants

Authors: Steven M. Grodsky, Leslie S. Saul-Gershenz, Kara A. Moore-O’Leary and Rebecca R. Hernandez

Reviewer: Matthew D. Trager

Summary:

The authors report results of a field-based study of queen butterfly oviposition on a potential host plant, Mojave milkweed, in the Mojave Desert. They found that the probability of oviposition increased with plant age and one measure of size and decreased throughout the field season. The authors discuss these results in the context of desert plant-insect interactions and conservation of the related monarch butterfly.

General comments:

The authors may have intended to study monarch butterflies, but since the surveys found zero monarch eggs I think it would improve the manuscript to refocus the introduction section on what was actually studied (i.e., the factors affecting queen butterfly oviposition on the Mojave milkweed, and perhaps what past studies have found or what theories predict). The current discussion section reviews some of the relevant background quite well, whereas the current introduction seems out of place since it sets expectations for the rest of the paper that simply are not met. There is probably a role for considering how the results compare with those for monarchs, or why monarch eggs were not found, but these should be primarily in the discussion and only briefly mentioned in the introduction.

The writing is quite good, and balances technical language with ecological storytelling in a way that is uncommon among peer-reviewed articles, though the title is a bit much for my taste. The scope of the study is relatively narrow (i.e., observational study conducted for one season on one life history stage), but the authors generally do not make inferences beyond what is warranted by the data. It would be interesting to know something about the broader context of the system, such as oviposition by queen and monarch butterflies across the entire suite of potential host plants, preference/performance relationships and spatial or temporal habitat partitioning. As is, this study is probably just barely enough for a paper.

Because I recommend such a major revision for the introduction and discussion, I have not made specific comments on those sections.

Specific comments (by line). Bold and italicized text is to be added, strikethrough for suggested minor word deletion.  

Ln 134 – I think a map of the study area may be helpful to understand the spatial scale of the study and aggregation of host plants/eggs. This could actually be a figure to support the results described on ln 193-195

Ln 171 – I want to know how many of the plants had eggs on them since individual plants had up to 19 (though I assume most had zero or 1 based on trying to interpret the y-axis values from Fig. 3). This could be a table or maybe just a single sentence

Ln 172-174 – the high abundance of host plants and eggs found in the early spring suggests that the surveys may not have been started early enough. There isn’t much that can be done about this, but it probably merits some discussion or at least a disclaimer that the earliest oviposition events may not have been within the study season.

Ln 185-189 – You already showed that oviposition was positively associated with plant canopy cover (ln 179-180) so an analysis of plant age/stage in which the ages are either defined by or strongly correlated with canopy cover really doesn’t make sense. In general, if you have continuous data (e.g., size measurements) then analysis based on those will be stronger than analysis based on continuous data that have been binned into arbitrary categories.

Fig 4. This should be a box/whisker plot or points with some measure of variation rather than a bar plot. Better yet, delete since Fig. 3 shows the same thing but better.

Round 2

Reviewer 2 Report

I appreciate the authors' thoughtful responses to my suggestions.  With respect to the framework of the paper, I continue to disagree with the emphasis on monarch ecology.  However, I consider this to be at the discretion of authors so I will not object further to the organization and content of the current introduction and discussion.

Including the map was helpful, but raised one additional question: were individual eggs used as data points for the analysis of spatial aggregation?  When I look the map it is hard to see the clustered effect that was found, and one potential reason could be that multiple eggs on plants (i.e., not independent samples) were considered as separate replicates.  Just something to make sure about, or disclose.  Oviposition data can be tricky, so authors should be very clear about their choices of analyses and their assumptions.

Fig. 4 (previously 3) clearly shows a non-normal (~Poisson, probably), zero-inflated distribution of egg counts.  That's typical and I think that most of the analysis deals with this common but challenging situation adequately.  However, Fig. 5 (previously 4) really doesn't fit the data.  I recognize that the point here is to show pairwise differences found with Tukey's test, but symmetrical measures of variability are misleading and bar plots are not very informative for count data.  I again recommend box and whisker plots because even though they probably will not demonstrate the statistical differences as clearly, they will show the distribution of data much more honestly.  I think the best way to simply deal with this kind of data is to conduct an initial analysis on presence/absence and then a second set of analyses testing predictors of egg abundance with the appropriate error distribution.

The changes made in response to the other reviewer improved the paper and I hope that my suggestions will be considered even though I am recommending that the paper be accepted.
